# Impact of a Single Virtual Reality Relaxation Session on Mental-Health Outcomes in Frontline Workers on Duty During the COVID-19 Pandemic: A Preliminary Study

**DOI:** 10.3390/healthcare13121434

**Published:** 2025-06-16

**Authors:** Sara Faria, Sílvia Monteiro Fonseca, António Marques, Cristina Queirós

**Affiliations:** 1Center for Psychology, Faculty of Psychology and Education Sciences, University of Porto (FPCEUP), 4200-135 Porto, Portugal; up201107842@edu.fpce.up.pt; 2School of Health, Polytechnic of Porto, 4200-072 Porto, Portugal; ajmarques@ess.ipp.pt

**Keywords:** frontline workers, virtual reality, stress reduction, anxiety, depression, resilience, COVID-19 pandemic, health workers, firefighters, police officers

## Abstract

**Background/Objectives**: The COVID-19 pandemic affected frontline workers’ mental health, including healthcare workers, firefighters, and police officers, increasing the need for effective interventions. This study focuses on the pandemic’s psychological impact, perceived stress, depression/anxiety symptoms, and resilience, examining if a brief virtual reality (VR)–based relaxation session could reduce psychological symptoms. **Methods**: In this preliminary study with data collected in 2025 from frontline workers who had served during the acute phase of the COVID-19 pandemic, 54 frontline workers completed a baseline assessment of the perceived psychological impact of COVID-19 pandemic, general perceived well-being, perceived stress (PSS-4), anxiety/depression (PHQ-4) and resilience (RS-25). Each participant then engaged in a 10-min immersive VR relaxation session featuring a calming 360° nature environment with audio guidance, after which questionnaires were re-administered. Paired samples *t*-tests and repeated-measures ANOVA evaluated pre-/post-session differences, and a hierarchical multiple linear regression model tested predictors of the change in stress. **Results**: Pre-session results showed moderate perceived stress and resilience and low depression/anxiety. Occupation groups varied in baseline stress, mostly reporting negative pandemic psychological effects. After VR, significantly perceived well-being increased, and stress decreased, whereas depression/anxiety changes were nonsignificant. Repeated-measures ANOVA revealed a main effect of time on stress (*p* = 0.003) without occupation-by-time interaction (*p* = 0.246), indicating all occupational groups benefited similarly from the VR session. Hierarchical regression indicated baseline depression and higher perceived pandemic-related harm independently predicted greater stress reduction, whereas resilience and baseline anxiety showed no statistically significant results. **Conclusions**: A single VR relaxation session lowered perceived stress among frontline workers, particularly those reporting higher baseline depression or pandemic-related burden. Limitations include the absence of a control group. Results support VR-based interventions as feasible, rapidly deployable tools for high-stress settings. Future research should assess longer-term outcomes, compare VR to alternative interventions, and consider multi-session protocols.

## 1. Introduction

The COVID-19 pandemic has precipitated a global mental health crisis. In the first year of the pandemic, the worldwide prevalence of anxiety and depression increased by an estimated 25% [1]. Frontline workers such as police, firefighters, emergency personnel, and HCWs have experienced high levels of psychological distress under the pandemic’s pressures and new demands [2,3,4,5,6,7,8,9,10]. Although each occupational group has diverse roles when responding to critical incidents [11], frontline workers are often the first to be present, which makes them a vulnerable group for psychological distress, as they face a high risk of exposure to stressful and traumatic events in their everyday jobs. Particularly for HCWs, several studies and systematic reviews indicated high levels of psychological distress among HCWs during the COVID-19 pandemic [9,12,13,14,15,16,17,18,19,20,21]. For example, Sialakis et al. [22] conducted a systematic review and found a prevalence of roughly 33.8% for depression and 41.3% for anxiety in HCWs during COVID-19, and the study by Son et al. [23] reported that between 23% and 67% of HCWs have clinically significant anxiety, 23–56% have depression, and 34–69% suffer from insomnia during the pandemic. The same psychological impact of the pandemic was related to police officers [24,25] and firefighters [26,27]. A recent EU-wide survey found that over 44% of workers reported an increase in work-related stress due to the pandemic, underscoring the persistent and widespread psychological impact of COVID-19 in the European workforce, especially in sectors such as healthcare and public safety [28].

It is important to note that even before the pandemic, these professionals were already at heightened risk of psychological distress due to their regular exposure to stressful and traumatic events in their daily work [6,29,30,31,32]. The pandemic, however, further exacerbated this risk. There are consequences for chronic stress and trauma exposure in frontline workers, and prolonged workplace stress and chronic stress can precipitate burnout, characterized by emotional exhaustion and other negative psychological outcomes. In fact, recent literature results showed that frontline workers had heightened burnout levels during the COVID-19 pandemic [2,5,15,33,34,35]. A study by Di Giuseppe et al. [15] reported that HCWs who were directly involved in care for COVID-19 patients reported more stress and burnout symptoms and more psychological distress than healthcare workers who were not exposed to the care for COVID-19 patients. In a large-scale Italian study by Trumello and colleagues [36] involving 627 healthcare workers, significantly higher levels of stress, burnout, secondary traumatic stress, anxiety, and depression were reported among professionals directly treating COVID-19 patients, especially those in high-contagion areas, and those working with COVID-19 patients were twice as likely to report an intention to seek psychological support compared to those who were not. During the pandemic, emergency workers have been exposed to the risk of much more immediate symptoms of depression, such as secondary trauma [37] or emotional exhaustion as a component of burnout [38]. These studies also considered how frontline workers were exposed to greater risks and developed more severe symptoms, which is in line with what had already happened during the SARS outbreak [39]. Additionally, chronic stress is associated with impaired cognitive functioning and mental disorders such as anxiety and depression [40,41]. High stress, burnout, and psychological distress not only endanger the well-being of the workers but can also impair job performance, threaten the quality of care, and contribute to negative workforce impacts, such as illness or turnover. Ensuring the mental well-being of frontline workers has, therefore, become an important priority, and international health agencies have urged that supporting frontline workers’ mental health is crucial [42,43]. It is important to note that some longitudinal studies indicate that frontline workers’ mental health trajectories differ markedly between the acute and recovery phases of the pandemic, especially after vaccination. A 42-month pan-European cohort found that depression peaked during the winter surge of 2020–2021 and then attenuated, although symptoms never returned to pre-pandemic baselines [44] For frontline workers, a two-year Iranian follow-up showed that the proportion of healthcare workers with severe emotional exhaustion rose from 22.9% to 31%, depersonalization from 7.4% to 10%, and perceived incompetence from 57.1% to 75.8%, even while COVID-specific anxiety itself declined [45]. Likewise, an eight-wave Canadian hospital survey (2020–2023) recorded steady increases in psychological distress and depersonalization, a decrease in personal accomplishment, and no late-phase recovery in emotional exhaustion despite lower case numbers and the formal declaration of the end of the pandemic [46]. Conversely, a large Chinese multi-center survey of HCWs showed that the prevalence of severe anxiety dropped from 52.3% during the surge (November–December 2022) to 8% seven months later [47]. Collectively, these longitudinal studies suggest that while certain COVID-19 fear-based symptoms may subside after vaccination and case declines, general distress and burnout can persist after the acute phases of the COVID-19 pandemic. The 2022 study by Laskawiec and colleagues [18] emphasizes that many experts predict a “post-pandemic stress disorder” and highlight that frontline healthcare professionals are at a particularly high risk of developing mental health disorders after the pandemic. In response, institutions and researchers have begun exploring interventions to mitigate stress and help promote the mental health of several professionals who faced high-pressure pandemic settings.

Virtual reality (VR)–based interventions have emerged as an innovative approach to stress management and mental health support [48,49,50,51,52,53,54,55,56,57,58], especially during and after the COVID-19 pandemic [59,60,61,62,63,64,65]. VR enables users to immerse themselves in computer-generated, interactive, immersive environments that can serve purposes such as relaxation, skills training, or therapeutic exposure [48,49,50,55,66,67,68].

A growing body of evidence suggests that VR experiences can produce measurable reductions in stress and anxiety. For example, Nijland et al. [61] found that a 10-min immersive VR relaxation session employed during a work break (using natural 360° environments and calming interactive exercises) led to an immediate 40% decrease in perceived stress among ICU nurses working with COVID-19 patients. Similarly, Beverly et al. [69] reported that an ultra-brief VR “tranquil scene” experience of about 3–5 min effectively reduced self-reported stress in frontline healthcare workers. These findings align with broader reviews indicating that most VR-based sessions provide and promote positive mental health benefits. In a recent systematic review by Li Pira et al. [70] of 20 studies of VR interventions, 65% of the studies showed beneficial effects of VR in reducing stress and negative emotional symptoms. VR has been successfully applied to treat or alleviate symptoms of anxiety disorders, depression, and post-traumatic stress by providing immersive relaxation, distraction from real-world stressors, or virtual exposure therapy in a controlled setting [70,71]. Taken together, these studies suggest that VR can rapidly induce psychological calm and reduce stress-related symptoms in diverse populations, including those under high workplace stress. VR interventions are also appealing in demanding contexts as the frontline workers have because they are relatively easy to deploy on-site, can be standardized and scaled, and offer an interactive escape that does not burden users’ concentration, which can sometimes happen with other traditional mindfulness exercises [61,72,73].

Despite the pandemic’s psychological and physical impact, many frontline workers demonstrate considerable resilience [46,74,75,76,77]. Research also shows that this resilience is heterogeneous [78,79] and can be overwhelmed by sustained adversity [80]. Resilience is commonly defined as the capacity to adapt positively and recover in the face of adversity or trauma [81,82]. Resilience can act as a protective buffer for mental health, helping individuals maintain or regain well-being during chronic stress [83,84]. For frontline HCWs, several studies seem to indicate that resilience has a protective effect against psychological distress: higher resilience has been associated with lower levels of anxiety, depression, and burnout symptoms during COVID-19 [85,86,87,88]. Along the same line, the study by Son et al. [23] found that resilience in HCWs mediated the impact of pandemic-related stress on their quality of life, such that those with greater resilience reported significantly better well-being despite high stress. A 2025 study by Giostra and colleagues [37] found, in a sample of 476 Italian Red Cross volunteers, that higher levels of secondary traumatic stress were strongly associated with burnout, but this association was significantly attenuated among volunteers reporting greater resilience skills. Similarly, a longitudinal study led by Bartone and colleagues [38] has shown that frontline workers who score high on dispositional resilience (e.g., hardiness) experience smaller increases in emotional exhaustion over time, even under sustained COVID-19 pandemic-related operational stress. The same seems to happen to frontline workers in general, as pointed out by the results of the study by Cheng and colleagues [89], which indicated that psychological resilience more effectively suppresses depressive and anxiety symptoms in frontline workers exposed to a traumatic event.

Lazarus and Folkman’s transactional model conceptualizes stress as a dynamic interaction between the individual and their environment, mediated by cognitive appraisal and coping processes [90]. When faced with a potential stressor, an individual first engages in primary appraisal, evaluating whether the situation poses a threat to their well-being. If the person judges an event as harmful or challenging, they then perform a secondary appraisal, which is an evaluation of available coping resources and options for dealing with the demand [90]. Stress is experienced if the perceived demands of the situation exceed the person’s perceived resources to cope [90]. In this framework, coping is defined as the constantly changing cognitive and behavioral efforts to manage specific external and/or internal demands that are appraised as taxing or exceeding one’s resources. An established distinction occurs between problem-focused coping (actively addressing or altering the source of stress) and emotion-focused coping (regulating one’s emotional response to the stressor), with emotion-focused coping more successful when stressors are not controllable and problem-focused coping more successful when stressors are controllable [91]. This theory can help understand that an intervention can reduce stress either by altering the stressor or by helping the individual cope with emotions. A brief VR relaxation experience does not remove the external stressor (e.g., job demands) but can help the individual manage the emotional and physiological response. Particularly, VR-based nature scenes have been shown to support these coping processes by quickly inducing positive affect and reducing acute distress (e.g., [92]. Complementing the transactional perspective, the Conservation of Resources (COR) Theory by Hobfoll [93] proposes that stress arises when people lose or anticipate losing personal, social, or material resources or when the investment of resources fails to produce sufficient return. Frontline workers’ repeated exposure to trauma, long shifts, moral dilemmas during COVID-19, and other factors are examples of cumulative resource loss (e.g., energy, sleep, sense of safety).

Stress Reduction Theory (SRT, [94,95]) and Attention Restoration Theory (ART, [96,97]) aim to explain why exposure to natural environments (where simulated environments can be included) can facilitate stress recovery. SRT, advanced by Ulrich [94,95], is a psycho-evolutionary model proposing that through evolution, humans are predisposed to respond positively to benign natural scenes (e.g., greenery, water) because such environments historically signaled safety and resources [98]. According to Ulrich, viewing an unthreatening natural environment triggers an initial automatic positive affective response (interest or calm) that rapidly reduces physiological arousal from stress [95]. ART, developed by Stephen and Rachel Kaplan [96] and Stephen Kaplan [97], focuses on the cognitive benefits of nature in restoring one’s capacity for directed attention, as contact with nature allows the directed attention mechanism to rest and replenish because natural environments are rich in soft fascination, absorbing involuntary attention through interesting stimuli (clouds, rustling leaves). In sum, Ulrich’s stress-reduction theory explains the immediate emotional and physiological relief people obtain from nature, while Kaplan’s restoration theory explains the later-stage recovery of cognitive capacity, and both are highly relevant to VR-based relaxation. A VR nature experience can simultaneously provide an instant calming effect (as per SRT’s affective pathway) and a sense of mental refreshment (as per ART’s attentional pathway). Evidence from recent studies supports the idea that simulated nature can invoke these restorative processes. For example, a study by Annerstedt and colleagues [99] found that a virtual forest scene with nature sounds led to faster physiological stress recovery (e.g., lower cortisol) after a stress task, comparable to real nature. Another example is a study by Yu and colleagues [100], which demonstrated that participants who experienced a 15-min interactive VR natural environment reported significant improvements in mood and reduced mental exhaustion.

Particularly for frontline workers, there is emerging empirical evidence in VR contexts. For example, Nijland and colleagues [61] deployed a 10-min VR relaxation app (“VRelax”) with COVID-19 ICU nurses during work breaks, observing approximately 40% drop in self-rated stress pre- vs. post-VR sessions, and nurses reported feeling happier and more in control after using VR. In another recent pilot, Beverly and colleagues [69] found that frontline healthcare workers’ stress scores fell significantly after just a 3-min 360° VR nature video: on average, from 5.5/10 before VR to 3.3/10 after, a large effect size (*d* = 1.1).

Considering the literature and the theoretical frameworks presented, we advance four hypotheses for this preliminary study:

**H1.** 
*A single VR-based relaxation session will significantly reduce perceived stress, anxiety, and depression and increase perceived well-being immediately after the session;*


**H2.** 
*The magnitude of stress reduction will not differ across occupational groups;*


**H3.** 
*Higher levels of perceived psychological harm due to the COVID-19 pandemic, anxiety, and depression will be associated with a greater reduction in perceived stress following the VR session;*


**H4.** 
*Higher baseline resilience will be associated with smaller reductions in perceived stress following the VR session.*


Therefore, the present preliminary study aims to achieve the following: (1) characterize the perceived COVID-19 pandemic’s psychological impact three years post-acute phase and the mental health (namely stress, depression, anxiety, and resilience) of frontline workers, namely HCWs, police, and firefighters; (2) examine whether a brief virtual reality (VR)–based relaxation session could reduce perceived stress, anxiety, and depression and promote mental health among frontline professionals; (3) assess occupational group differences in stress reduction and (4) identify whether baseline characteristics (perceived pandemic harm, anxiety, depression, and resilience) are associated with post-session stress reduction.

## 2. Materials and Methods

### 2.1. Participants

A convenience sample of 54 Portuguese frontline workers (13 healthcare workers, 13 firefighters, and 28 police) with an average age of 43.85 years (median = 45, SD = 9.16, ranging from 21 to 59 years) was recruited between January and March 2025. Because participants were asked to recall the acute phase of the COVID-19 pandemic (2020–2022), this aspect of the study is retrospective. Although that phase had ended three years earlier, the VR session specifically targeted the residual and chronic stress that can persist long after the crisis; this temporal distance is acknowledged as a study limitation. Participants were purposively recruited through the team’s institutional contacts in Portugal, institutions with which the research team had previously collaborated. Invitations were delivered in person or by e-mail, outlining the study’s aims and giving recipients the option either to contact the research team directly or to leave their e-mail address to ensure that a session could be scheduled. A snowball sampling procedure was then employed, enabling anyone who wished to participate to engage. We first e-mailed 11 institutional contacts (4 from health institutions, 4 police, 3 fire stations) with an invitation letter describing the study. Those contacts forwarded the invitation to their colleagues. Fifty-four individuals expressed interest, met all criteria, and volunteered. The inclusion criteria were: (1) having been on duty during the acute phase of the COVID-19 pandemic (2020–2022), (2) being at least 18 years old, and (3) having provided informed consent. The only exclusion criterion was the presence of any health contraindications, specifically photosensitive epilepsy, motion sickness, neurological conditions (such as multiple sclerosis or brain injuries), or balance disorders (including labyrinthitis). Of the total participants, 29.6% were female (*n* = 16) and 70.4% were male (*n* = 38). For confidentiality and ethical reasons, gender is not reported separately within each occupational group. Nonetheless, the pattern in our sample represents national figures as most police officers and firefighters were male, whereas most healthcare workers were female, consistent with official statistics for these professions in Portugal [101,102,103]. The educational qualifications varied, with 31% having compulsory education, 13% having a bachelor’s degree, and 29.6% having a master’s degree. Most of the participants were in a stable relationship (79.6%) and had children (68.5%). Regarding their profession, the majority were not displaced (75.9%) and had 18.7 years of service (M = 18.70, SD = 10.19). Due to confidentiality reasons, the participants’ areas of work will not be stated, but the majority worked in the North of Portugal (92.6%).

### 2.2. Procedures

This study adhered to the principles declared in the 1964 Declaration of Helsinki and received the approval of the Ethics Committee. All the participants provided their informed consent before taking part in the study, and participation was voluntary, anonymous, and confidential.

Before primary data collection, a pilot study was conducted to ensure the clarity and suitability of all questionnaire items. This preliminary phase involved both the research team and a small sample of participants, allowing for refinements that enhanced the instrument’s comprehensibility for the target population.

An online survey was developed in Google Forms to recruit participants at the end of 2024. Once inclusion criteria were verified, participants scheduled their VR sessions and completed all questionnaires on the online platform SOMSII Innovation & Research–FlexSaúde platform, which allows immediate feedback after submitting the results, which was preferable for the specific aims of the project. This platform was co-created with the help of most of the authors and provides an easy integration with IBM-SPSS, minimizing transcription errors.

The study used a non-probabilistic snowball sampling approach. Although this sampling method can limit randomization (due to reliance on volunteer participation), it also facilitates efficient data collection and can increase sample diversity [104]. Through established protocols and institutional networks, the combined survey and VR session were disseminated to recruit participants, who could either come to the university or receive a visit from a researcher at their own institution. To exclude a location effect, an independent-sample t-test was performed, comparing the session location (university and institutions) on every study variable, and all comparisons were nonsignificant. We then fitted a 2 (Time: pre-session vs. post-session) × 2 (session location: university vs. institution) repeated-measures GLM on PSS-4 and PHQ-4 scores (assumptions met), which showed no main or interaction effect of location. Because neither baseline levels nor the magnitude of stress, anxiety, and depression differed by session location, this variable was considered non-influential and was not included in subsequent models.

Regarding the VR session, each participant’s overall session lasted approximately 40 min, considering the following protocol: ~15 min to complete the pre-session questionnaire, ~15 min for the VR session, ~5 min to answer the post-session questionnaire, and ~5 min for debriefing. The VR sessions took place in a quiet, private room at either the research team’s university or the participant’s institution. The session format was as follows: the VR session was non-interactive (passive 360°), requiring no user actions beyond looking around the virtual environment; participants were instructed to remain seated comfortably and to practice mindful attention to the immersive scenario, taking deep breaths and following the scenario audio instructions; if any negative reaction (e.g., cybersickness) arose, participants were free to remove the headset at any time and completely stop the session. At the start of the session, participants were asked about potential health contraindications based on the VR headset’s user manual recommendations (as literature is lacking or conflicting regarding explicit contraindications for VR; see also [105,106,107,108]. A standalone virtual reality headset (VR SHINECON headset) was used to deliver the VR session, which is easy to handle. Each participant received brief instructions on how to use the device and was monitored by a researcher throughout the session to ensure comfort and safety. Proper headset fit was checked, and participants were reminded they could stop anytime if discomfort arose. No interruptions or severe side effects (e.g., severe cybersickness) occurred. At the end of the session, the headset was removed, and the participant remained seated for one minute to readjust to the real environment before starting the post-session questionnaire. Finally, the participants’ debriefing included talking with the participants about their experience and mental and physical condition after exposure, ensuring the participant’s safety and well-being after the session.

No separate control group (e.g., a non-intervention or alternative-intervention group) was included due to practical constraints, hard access to this sample, and the preliminary nature of this initial trial. There were no dropouts since the session and assessment were completed in one session.

### 2.3. Instruments

A two-section (baseline assessment or pre-VR session and post-VR session) online questionnaire was conducted. The pre-VR session questionnaire collected socio-demographics, the perceived psychological impact of the COVID-19 pandemic, a single-item rating of general perceived well-being, and various questionnaires designed to evaluate stress (Perceived Stress Scale-4, PSS-4), depression and anxiety (Patient Health Questionnaire, PHQ-4) and the Resilience Scale (RS-25). The post-VR session questionnaire measured PSS-4, PHQ-4, and general perceived well-being. Identical measures (the PSS-4, PHQ-4) were administered immediately before and after the VR session, specifically to capture acute, session-specific psychological effects. Broader or global psychological health measures served as supplementary assessments but were not designed to detect short-term fluctuations directly attributable to the brief VR exposure. Our aim was a preliminary evaluation of immediate intervention effects, providing justification for future comprehensive studies. It is noted that for the post-VR session, the application of the PHQ-4 only began later due to changes in the main aims of the project, which means there is a difference in the number of participants in the pre- and post-VR sessions for PHQ-4, which is clearly reported in the Results section.

To characterize how frontline workers experienced and lived through the COVID-19 pandemic, a set of 11 questions was gathered after discussion with field professionals and relevant literature. This section was developed to seek to understand the impact that the COVID-19 pandemic had on the psychological and emotional health of professionals, both in retrospect during the most intense period of the pandemic and during the data collection period. The section included questions about how professionals self-assessed their current psychological health status (Likert scale, 0 = very bad to 4 = very good), compared with the most acute phase of the COVID-19 pandemic (2020–2022), whether they felt that their psychological health status had improved or worsened (multiple choice; improved, worsened, unchanged) and a question that sought to understand to what extent the professional considered that what they experienced during the COVID-19 pandemic had harmed their psychological well-being (Likert scale; 0 = did not harm at all to 5 = harmed a lot). A group of multiple choice questions (yes, no) was designed to understand whether, during the most acute phase of the pandemic, these professionals requested psychological help, were infected, had direct or indirect contact with people infected with COVID-19 both in the exercise of their work and in their family, had deaths due to COVID-19 either of colleagues or acquaintances at work or users/patients, and, finally, whether they considered that they had ever been the target of stigma/discrimination in their work.

The Perceived Stress Scale (PSS-4; [109], Portuguese version [110]) was used to measure stress. This instrument assesses the frequency of perceived stress by four items on a five-point Likert scale ranging from 0 (never) to 4 (very often) and results in an overall scale (α = 0.83, four items). Two items are reverse-worded. Higher scores indicate higher perceived stress. To measure anxiety and depression, the Patient Health Questionnaire (PHQ-4; [111], Portuguese version [112]) was used. This instrument assesses four items on a four-point Likert scale ranging from 0 (not at all) to 3 (nearly every day), and each pair of items measures the frequency of anxiety (α = 0.89, two items) and depression (α = 0.80, two items). Higher scores indicate higher anxiety and depression, respectively. It is noted that the PHQ-4 is validated for screening anxiety and depression in occupational samples, the general population, and large-scale surveys (e.g., [113,114,115,116,117]). To measure resilience, the Resilience Scale (RS-25; [82], Portuguese version [118]) was used. This scale assesses 25 items on a seven-point Likert scale ranging from 1 (totally disagree) to 7 (totally agree) and results in an overall scale (α = 0.85, 25 items). The RS-25 produces a total with higher scores reflecting higher resilience.

For the VR relaxation session, each participant engaged in a single 10-min immersive scenario featuring a calming 360° nature environment with audio guidance (“The Secret Garden”), developed by Riva and colleagues [64], which was translated and adapted to the Portuguese language and context by the research team. This immersive scenario was originally developed by Riva and collaborators as part of a VR self-help protocol called “COVID Feel Good,” aimed at reducing stress and promoting well-being during the pandemic. In the “Secret Garden” experience, the participant is immersed in a garden featuring vegetation, flowers, trees, a river, and other relaxing natural elements, including natural sounds, to enhance the sense of presence and relaxation [56].

### 2.4. Data and Statistical Analysis

Descriptive statistics, namely the means and standard deviations, were used to describe continuous variables, and frequencies and percentages were used to describe categorical variables. The study of this normality was carried out using the Kolmogorov–Smirnov test and the asymmetry (<|3|) and kurtosis (<|8|) criteria of Kline [119]; no significant outliers were detected (*Z* < |3.5|; [119]). Internal consistency of the instruments was examined considering Cronbach’s Alpha and McDonald’s Omega. Following Nunnally’s [120], alpha (α) values are considered adequate when above 0.80.

Paired samples t-tests assessed pre-VR session with post-VR session changes, a repeated-measures ANOVA tested differences by occupation group, and a hierarchical multiple linear regression model (all assumptions were assured; Durbin–Watson = 1.95) examined whether baseline depression, anxiety, resilience, and perceived pandemic impact predicted greater post-session stress reduction. Effect sizes for paired differences were calculated using Cohen’s d for repeated measures (within subjects), with the mean of the differences divided by the standard deviation of the differences [121]. In order to perform the hierarchical regression on stress reduction, we calculated a difference score [122] (hereafter called delta_stress) by subtracting the post-session stress measure from the pre-session measure, such that higher delta_stress values reflect a greater decrease in perceived stress. Dichotomous variables were transformed into dummy variables [123]. All the statistical analyses were carried out on IBM SPSS software, version 30.0 for MacOS, and were interpreted with the assumption of a significance level of 5% (two-tailed).

## 3. Results

### 3.1. Characterization of the Psychological Impact of the Pandemic

Descriptive statistics and frequencies were analyzed to characterize participants’ responses related to the psychological impact of the COVID-19 pandemic, as shown in Table 1. Participants, on average, rated their present mental health at 2.54 (*SD* = 0.79) on a 1–4 scale, suggesting moderate self-assessed well-being. By contrast, the mean response (*M* = 1.63, *SD* = 1.45) regarding how severely the pandemic harmed their psychological health (0–5 scale) indicates that many felt at least somewhat affected, though severity varied considerably among individuals. When asked whether their mental health had improved or worsened compared to the peak pandemic phase, 22.2% reported improvement, 27.8% reported deterioration, and half reported no change in their mental health. It is noted that only 3.7% sought professional psychological help despite the stressors of the pandemic and despite 13% percent reporting having a colleague or acquaintance die from the virus, and 44.4% having direct professional contact with individuals who passed away from COVID-19. Furthermore, 16.7% felt they were subjected to stigma or discrimination at work related to COVID-19.

### 3.2. Current General Psychological Health, Stress, Anxiety, Depression, and Resilience: Pre- and Post-VR Session Descriptive Measures

The descriptive statistics for general current psychological health, stress, anxiety, and depression pre- and post-VR session scores and resilience pre-VR session scores are shown in Table 2. The values of skewness and kurtosis were within the normal ranges.

### 3.3. Current General Psychological Health, Stress, Anxiety, Depression, and Resilience: Pre- and Post-VR Session Paired t-Test Results

To evaluate the immediate impact of the VR session on general psychological health, stress, anxiety, and depression, we conducted paired-sample t-tests comparing pre-VR vs. post-VR scores on the general psychological health item PSS-4 and PHQ-4 (Table 3). This analysis comprises the entire sample as one group, testing whether there was a significant mean change following the VR session. Effect sizes were calculated using Cohen’s *d*. Paired-sample tests revealed a significant increase in perceived well-being (mean difference = 0.14, *t*(53) = *−*2.57, *p* = 0.013) and a reduction in PSS (mean difference = 0.14, *t*(53) = 2.43, *p* = 0.018) after VR across groups, whereas PHQ changes were not statistically significant.

### 3.4. Stress Reduction on the VR Session and Occupational Groups: Repeated-Measures ANOVA Results

To examine whether the degree of improvement in stress levels differed between the three occupational groups (police, HCWs, and firefighters), a repeated-measures ANOVA on PSS-4 scores was performed. The ANOVA results revealed a significant main effect of Time (*F*[1,50] = 4.80, *p* = 0.033, partial *η*^2^ = 0.086), confirming the overall reduction in stress from pre- to post-VR session (as already shown by the *t*-tests). The main effect of Occupation was also significant (*F*[2,50] = 8.81, *p* < 0.001, partial *η*^2^ = 0.257), with firefighters reporting higher average stress than police (mean difference = 0.93, *p* < 0.001) and HCWs (mean difference = 0.83, *p* = 0.006), and no other pairwise differences were significant. However, everyone benefits in a similar way. This main effect essentially reflects the baseline difference between the groups, which persisted to some extent post-session (though both groups improved). Lastly, the Time × Occupation interaction was not significant (*F*[2,51) = 0.04, *ns*). This indicates that the pattern of stress reduction was statistically parallel across all occupational groups, and they did not differ statistically significantly in terms of the change in stress from pre- to post-VR session. This means that all groups had a similar change from pre- to post-VR session, and no group improved more than the others.

### 3.5. Predictors of Post-Session Stress: Hierarchical Linear Regression Results

Hierarchical linear regression was performed to examine the contribution of the perceived psychological pandemic harm and anxiety, depression, and resilience on frontline rescuers’ pre- and post-VR session stress levels. Dimensions of the perceived psychological pandemic harm entered the first block of the regression analysis, and anxiety, depression, and resilience entered the second block. It is noted that for the dependent variable, we calculated a difference score (called variable delta_stress) by subtracting the post-session stress measure from the pre-session measure, such that higher delta_stress values reflect a greater decrease in perceived stress, and this was the dependent variable used in this analysis. The two regression models are presented in Table 4.

In the first model, the perceived psychological pandemic harm significantly explained 9% of the variance of the post-session stress reduction (*F*[1, 53] = 6.256, *p =* 0.016). That is, participants who reported greater psychological harm from the pandemic tended to have a greater reduction in stress after the VR session. In the second model, when adding depression, the explained variance of the post-session stress reduction increased to 12.1% (*F*[4, 53] = 2.830, *p* = 0.034). These effects are considered small-to-medium by Cohen [121]. Here, only depression positively explained the post-session stress reduction (*β* = 0.42, *p* = 0.035). The variables of anxiety and resilience did not survive in this model.

## 4. Discussion

The COVID-19 pandemic has had a profound psychological impact on frontline workers, including healthcare workers, police officers, and firefighters. The results of the present study on the psychological impact of the pandemic showed that participants reported a moderate level of current psychological health. By contrast, the mean response (*M* = 1.63, *SD* = 1.45) regarding how severely the pandemic harmed their psychological health (0–5 scale) indicates that many felt at least somewhat affected, though severity varied considerably among individuals. When asked whether their mental health had improved or worsened compared to the peak pandemic phase, 22.2% reported improvement, 27.8% reported deterioration, and half reported no change in their mental health. Although this suggests that some frontline workers seem to have coped effectively, the fact that over a quarter perceived ongoing or increased distress aligns with recent findings suggesting that many rescuers and frontline workers continue to experience persistent psychological distress beyond the initial crisis phase [3,5,6,18,124,125,126]. Although participants indicated that their well-being was affected by pandemic stressors, only 3.7% sought professional psychological support, which may reflect the reluctance to engage in mental health care, which recent literature attributes to stigma, logistical barriers, or underestimation of personal distress [7,9,124]. In this study, 16.7% of participants reported that they felt stigma or discrimination within the workplace. Also, 13% of participants reported having a colleague who passed away from COVID-19, and 44.4% experienced direct contact with dying patients or clients, a degree of exposure that other studies have linked to a heightened risk for chronic stress, anxiety, depression, trauma-related symptoms and moral distress in frontline workers [3,4,15]. These findings suggest that although some individuals noted improvement in psychological health post-peak-pandemic, a significant proportion remains vulnerable to ongoing stressors and has not pursued sufficient support, aligning with other findings on post-pandemic mental health among frontline workers. It is important to note that self-reported help-seeking and retrospective appraisals—as was the case in this section of the questionnaire where participants were asked to recall the most intense period of the COVID-19 pandemic—may be underestimated due to occupational stigma [127] and a minimization of the psychological state—as avoidance and cognitive suppression—have been documented as coping strategies in frontline workers/medical responders [128,129].

Regarding the mental health of the participants, pre-session scores indicated moderate levels of perceived stress (*M* = 1.15, *SD* = 0.80 on a 0–4 scale), low to moderate anxiety (*M* = 0.57, *SD* = 0.74 on a 0–3 scale), and low depression scores (*M* = 0.46, *SD* = 0.70), along with a relatively high average resilience score (*M* = 5.76, *SD* = 0.52 on a 1–7 scale). These findings suggest that while participants were not reporting severe psychological distress at the time of assessment, stress was still a concern. This is consistent with other recent research highlighting the psychological impacts among frontline workers even in the post-peak phase of the pandemic [3,5,6,18,124,125,126]. The moderate stress and low depression results in this study assessed three years after the crisis peak, may be due to the post-vaccination, as also reported in the recent 2025 European general-population study by Zrnić Novaković and colleagues [44] and the stabilized, but still relevant distress seen in frontline HCW’s [46,47]. The moderate levels of stress reported are in line with studies showing that rescuers often experience cumulative stress exposure after the acute phase of public health crises has passed [18,34]. Furthermore, while the average anxiety and depression scores appear low, at least one participant reported the maximum value. Taken together, both the study’s initial assessments and the literature highlight frontline workers endured a decline in mental well-being during and after the pandemic, reinforcing the need for interventions to mitigate this “silent pandemic” of psychological distress.

The study also evaluated single-session VR relaxation as a rapid support measure to promote mental health among frontline workers. The results showed improvements from pre- to post-session in some of the participants’ mental health indicators, with participants reporting lower stress and higher current psychological well-being levels, although anxiety and stress were not statistically significant. These findings are consistent with a growing body of literature demonstrating that even a short VR-based relaxation or mindfulness experience can induce significant stress relief in frontline workers [61,69]. This pattern is consistent with SRT [94,95] and ART [96,97], as SRT posits that viewing benign natural scenes triggers an automatic positive-affect response that quickly lowers arousal, and ART suggests that such scenes capture involuntary attention, allowing directed attention to recover, which can explain the immediate decreased in stress levels. It is important to note that although baseline distress was lower than early-pandemic figures, the brief VR session still produced a significant incremental reduction in stress levels. This implies that immersive relaxation retains relevance even after the acute phases of the crisis have passed and can be used to alleviate distress in several crises, future pandemics, and day-to-day life stressors faced by frontline workers. Nevertheless, future studies should verify whether comparable or larger effects emerge when baseline distress is higher, such as during new crisis/pandemic surges or in regions with low vaccine coverage. The present study aligns with findings from Vagni and colleagues [130], who emphasized the effectiveness of the Stopping Negative Thoughts and Emotions strategy for mitigating emotional exhaustion among frontline workers during COVID-19. This approach enables individuals to shift attention away from distressing imagery and intrusive negative thoughts associated with traumatic work experiences [37,38,130]. Similarly, VR relaxation sessions can provide a structured, immersive relaxation experience that can help redirect attention to calming virtual environments, which can temporarily help stop the cycle of distressing thoughts and emotions. Our findings reinforce that VR-based interventions can serve as an accessible and practical tool for frontline workers to enhance positive coping mechanisms directly related to stress management during crisis periods, potentially buffering the negative psychological effects associated with frontline work. The benefits of the VR session were achieved with minimal time investment, which shows as a critical advantage in high-stress occupations where workers often have little free time, but it is important to consider that these immediate post-session gains may represent short-term relief, and the need for follow-ups or repeated sessions are needed.

One objective of the study was to determine if the impact of VR sessions differed among occupational subgroups (healthcare workers, police, and firefighters). The results suggested that all three professional groups experienced stress reduction from the VR sessions. This seems to suggest that the relaxation session appeared to be broadly applicable across these distinct frontline workers. This is also aligned with the study of Beverly et al. [69] that focused on healthcare staff, which reported that stress relief from a VR nature simulation did not significantly differ by provider type (nurses, physicians, support staff, etc.), reinforcing the session was equally helpful to various roles. This result seems promising, as it appears to indicate the tool’s versatility, an important consideration for mental health interventions intended for widespread use. Also, this outcome underscores that VR relaxation is flexible, as it can be delivered in a hospital break room, a police station, or a firehouse with similarly positive outcomes.

Lastly, we provided a hierarchical regression model aimed at understanding how baseline characteristics—specifically, perceived pandemic-related harm, initial anxiety/depression levels, and resilience—influenced participants’ responses to the VR session post-stress reduction. In the first model, perceived psychological harm from the pandemic emerged as a significant predictor of greater stress reduction following the session (*β* = 0.328, *p* = 0.016), explaining 9% of the variance. This suggests that those who felt more impacted by the pandemic experienced greater benefits from the VR session, possibly due to a higher need for psychological relief. This finding is consistent with recent literature indicating that digital interventions may be especially effective for individuals experiencing elevated distress, as immersive experiences can offer an accessible buffer against emotional overload [50,65]. In the second model, depression was the only significant additional predictor (*β* = 0.415, *p* = 0.035), further increasing the explained variance to 12.1%, which meant that individuals with higher baseline depression showed greater reductions in stress, which may reflect a stronger contrast effect or heightened susceptibility to short-term affective shifts following immersive interventions. These results also align with studies that found that VR relaxation may disrupt ruminative patterns and elicit calming physiological responses in mildly to moderately depressed individuals [70,71,131]. Neither baseline anxiety nor resilience significantly predicted stress reduction; this might be due to the short session duration or the ceiling effects on resilience (which was already high in the sample). For anxiety, this may highlight the need for stratified care, where frontline workers already suffering from anxiety might benefit from additional resources, such as counseling or psychiatric, with VR relaxation as an adjunct rather than an only strategy. For resilience, this aligns with the idea that while resilience buffers against distress, it may not significantly modulate immediate responsiveness to brief interventions, especially if individuals already possess effective coping mechanisms [89]. Alternatively, the non-significance of resilience could point to the limits of trait-based predictors in explaining the short-term change, particularly when that change is driven more by emotional intensity—as in the case of pandemic harm—than by stable, protective characteristics. This finding resonates with broader discussion in the literature about the predictive limits of trait-based resilience measures, as many resilience scales conceptualize resilience as a stable disposition, correlating negatively with distress but falling short in predicting dynamic patterns of change in mental health [83,132]. The hierarchical regression aimed to provide further insight when interpreted through COR theory [93], which states that stress intensifies when individuals lose—or expect to lose—valuable resources and that resource gains are especially powerful for those already depleted. Participants who appraised the pandemic as especially harmful or who entered the study with higher depressive symptoms had the highest levels of post-session stress reduction. This demonstrates that the VR experience acted as a short-term resource gain that partially replenished depleted emotional reserves. Overall, these findings are valuable for tailoring future interventions, as they suggest screening for baseline mental health could identify who might need a longer session or supplemental support.

This study has some limitations: the results may be influenced by the “myth of the healthy worker” [133], since those who are already in a worse mental state have less mental availability to fill out surveys or may even be on sick leave; this was a single-session application with measurements taken immediately before and after the VR experience, used to understand short-term effects and, therefore, a sustained impact or whether participants’ baseline mental health eventually returned to pre-session levels could not be assessed; sample size and generalizability are additional considerations, as the study has a relatively small sample, and this can reduce statistical power (meaning some effects or differences might go undetected-type II error) and the findings should be generalized with caution; another limitation is the reliance on self-report measures, as all outcomes were measured via questionnaires, which are subjective; the study lacked a control group; although the acute phase of the COVID-19 pandemic ended approximately three years before data collection, the VR session was deliberately aimed at the residual, and often chronic stress that persists well beyond the crisis, as documented in several recent studies-this temporal distance may reduce ecological validity and is acknowledged as a limitation; perceived pandemic harm was assessed with a single item and, although single-item measures can limit reliability, they reduce respondent burden in brief protocols and have shown acceptable validity for clear, specific constructs (e.g., [134,135]).

This preliminary study was intended to determine whether further experimental designs, such as randomized controlled trials with extended follow-ups, would be justified and meaningful for future studies, as it showed promising results in frontline workers who were on duty during the COVID-19 pandemic in Portugal. Future directions should include controlled trials, larger and more diverse samples, follow-ups, and the integration of objective metrics (such as physiological indicators of stress like heart rate, blood pressure, cortisol levels, etc). Moreover, longitudinal studies are needed to assess the durability of VR’s effects on mental health indicators.

Practical implications. The present findings support the integration of brief, self-guided VR “micro-breaks” into routine occupational health provisions for frontline workers. Portable, cost-efficient head-mounted displays can be stationed in hospital rest areas, police break rooms, fire stations, and other institutions with frontline workers, enabling them to initiate a 10-min immersive relaxation session without staff supervision. These interventions would provide frontline workers with accessible tools to momentarily detach from persistent negative thoughts, emotional exhaustion, and secondary traumatic stressors that commonly arise during emergency situations. At the organizational level, such VR interventions can serve as a strategy to respond to cumulative resource loss and enhance positive coping strategies and resilience, with the potential to attenuate psychological distress, reduce sickness absence, and improve workforce sustainability in high-pressure settings.

## 5. Conclusions

This study contributes to the growing body of literature demonstrating that virtual reality (VR)–based relaxation can offer immediate psychological relief to frontline workers, including HCWs, police officers, and firefighters, many of whom continue to experience psychological distress years after the peaks of the COVID-19 pandemic. The findings suggest that a brief, portable, rapidly deployed tool with VR can lead to significant improvements in self-reported stress and current psychological well-being, helping to promote mental health in high-stress occupations. Importantly, these benefits were observed across occupational groups, highlighting the tool’s flexibility and potential for broad implementation in frontline settings. Moreover, the identification of baseline depression and perceived pandemic-related harm as predictors of greater stress reduction supports the need for stratified care models in which low-intensity interventions like VR can be optimized and targeted for those most likely to benefit.

Caring for caregivers and protectors is not only a moral duty, but it is also a way to strengthen healthcare, security, and emergency response systems. The experiences of COVID-19 brought awareness that mental health support is as essential as any other preparedness measure, and adopting tools like VR relaxation can be a way to promote mental health for those who serve on the frontlines.

## Figures and Tables

**Table 1 healthcare-13-01434-t001:** Descriptive Statistics and Frequencies Related to the Psychological Impact of the Pandemic.

Variable	Total (*n* = 54)		
	Min, Max.	*M*	*SD*
How would you generally rate your current psychological health? ^1^	1, 4	2.54	0.79
To what extent do you consider that your experiences during the COVID-19 pandemic (2020–2022) harmed your psychological well-being? ^2^	0, 5	1.63	1.45
		** *n* **	**%**
Compared to the most intense period of the COVID-19 pandemic (2020–2022), do you feel your psychological health has improved or worsened?			
Improved		12	22.2
Worsened		15	27.8
Unchanged		27	50
During or shortly after the COVID-19 pandemic (2020–2022), did you request psychological help?			
Yes		2	3.7
No		52	96.3
Did anyone at your workplace (colleagues or acquaintances) pass away from COVID-19 between 2020 and 2022?			
Yes		7	13
No		47	87
Between 2020–2022, in the course of your work, did you have contact with patients/clients (or in other professions, users of your service, etc.) who passed away from COVID-19?			
Yes		24	44.4
No		30	55.6
Between 2020–2022, do you believe you were subjected to stigma/discrimination at work because of COVID-19?			
Yes		9	16.7
No		45	83.3

*Notes:* ^1^ Range: 1 (bad), 4 (very good); ^2^ Range: 0 (did not harm at all), 5 (harmed a lot).

**Table 2 healthcare-13-01434-t002:** Descriptive Statistics for the Levels of Stress, Anxiety, Depression, and Resilience.

Variable		*n*	*M*	*SD*	Skewness	Kurtosis
How would you generally rate your current psychological health? (1–4)	Pre	54	2.54	0.79	0.090	−0.164
	Post	54	2.76	0.91	−0.101	−1.089
Stress (0–4)	Pre	54	1.15	0.80	0.576	0.004
	Post	54	1.01	0.72	0.309	−1.037
Anxiety (0–3)	Pre	54	0.57	0.74	1.204	0.839
	Post	37	0.49	0.72	1.867	3.901
Depression	Pre	54	0.46	0.7	1.168	−0.016
	Post	37	0.32	0.57	1.956	4.489
Resilience (1–7)	Pre	54	5.76	0.52	−0.603	0.541

**Table 3 healthcare-13-01434-t003:** Results of the Paired Samples *t*-Test.

	Paired Samples (Pre- vs. Post-VR)				
	*n*	*M* Δ	95% CI for Δ	*T (df)*	*d* ^1^
How would you generally rate your current psychological health? (1–4)	54	−0.22	[−0.40, −0.05]	−2.57 (53) *	0.63
Stress (0–4)	54	0.15	[0.03, 0.26]	2.434(53) *	0.43
Anxiety (0–3)	37	−0.1	[−0.27, 0.08]	−1.13 (36), *ns*	0.51
Depression	37	−0.03	[−0.19, 0.14]	−0.33 (36), *ns*	0.5

*Notes*: ^1^ Effect size (Cohen, 1988 [121]): *d* ≤ 0.2-small; *d* = ]0.2, 0.5]-moderate; *d* = ]0.5, 1.0]-large; *d* > 1.0-very large. * *p* < 0.050, *ns* = non-significant.

**Table 4 healthcare-13-01434-t004:** Post-Session Stress (Delta_Stress) Predictors: Perceived Impact of the Pandemic on Well-Being, Anxiety, Depression and Resilience.

	*B*	*SE B*	*β*	*p*
Model 1				
Perceived psychological pandemic harm	0.098	0.039	0.328	0.016
*F*(1,53)	6.256 *			
*R* ^2^ *a*	0.090			
Model 2				
Perceived psychological pandemic harm	0.093	0.039	0.312	0.022
Anxiety	−0.178	0.114	−0.302	0.124
Depression	0.257	0.119	0.415	0.035
Resilience	0.036	0.110	0.043	0.741
*F*(4,53)	2.830 *			
*R* ^2^ *a*	0.121			

* *p* < 0.05.

## Data Availability

The datasets presented in the article are not readily available because the data are part of an ongoing study.

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
