# Peer review of "Impact of a Single Virtual Reality Relaxation Session on Mental-Health Outcomes in Frontline Workers on Duty During the COVID-19 Pandemic: A Preliminary Study"

_healthcare, 2025, doi:10.3390/healthcare13121434_

Round 1

Reviewer 1 Report

Comments and Suggestions for Authors

The authors present a study entitled “Mental Health and the Impact of COVID-19 Pandemic in Healthcare Workers and other Frontline Workers: a Single VR Relaxation Session.”

Please, consider revising the title to more clearly reflect the study's focus and methodology—for example, by emphasizing the VR intervention and its effects—so readers can immediately grasp the paper's specific contribution to understanding mental health support for frontline workers during COVID-19.

Abstract

  • Although the retrospective nature of the data collection, which took place in 2025 well after the acute phase of the COVID-19 pandemic, is noted in the methodology section, this important contextual detail should also be clearly stated in the abstract to prevent any potential misinterpretation of the study's temporal relevance.
  • It is recommended to insert spaces where appropriate between words and abbreviations, for example changing 'stress(PSS-4)' to 'stress (PSS-4)'.
  • An important limitation of the study, which is already evident from the abstract, is the absence of a control group. This should be explicitly acknowledged and discussed, both in the abstract and in the main text, to provide a balanced interpretation of the findings.
  • The abstract lists the statistical analyses performed, but it does not clearly state the hypotheses being tested or the rationale behind the choice of these specific analyses. Including this information would help readers better understand the study's objectives and the logic guiding the analytical approach. In particular, in my opinion, the rationale for testing the hierarchical regression model is unclear from the abstract. The absence of a clearly stated hypothesis or theoretical justification makes it difficult to understand the logic behind this analytical choice. I will return to this point in the discussion of the methodology and results.

Introduction

  • The authors state that 'Despite the pandemic’s psychological and physical impact on frontline workers, these professionals are recognized as resilient' (67-69). However, in my opinion, it is not entirely accurate to claim that all frontline workers are resilient. Additionally, the role of resilience is not explicitly addressed within the introduction, and it would be beneficial to clarify its significance and how it relates to the study's focus.
  • There is a notable absence of a theoretical framework to guide the study, both in terms of stress and VR. It is essential to emphasize the importance of having a well-established theoretical framework in these areas to provide a solid foundation for the study. Without such a framework, the interpretation of the findings may lack context and theoretical grounding, which could undermine the study's scientific rigor.
  • I do not understand the logic behind the fourth aim outlined by the authors, particularly how the baseline is expected to reflect a reduction in stress.
  • Additionally, a significant limitation of the study is the lack of clearly formulated hypotheses, which makes it difficult to understand the specific predictions and expectations guiding the research.

Materials and methods

  • The phrase 'The participants were purposely chosen through the team’s protocols contacts in Portugal' is unclear. It is important to specify how many participants were selected and the exact process of selection. Additionally, how many individuals were initially contacted should be stated to provide clarity on the sampling procedure.
  • An important limitation that emerges from the methodology is that we are far removed from the COVID-19 pandemic, as already noted in the abstract. Furthermore, the sample size for each group is very small, which raises concerns about the statistical power and generalizability of the findings.
  • It would be important to specify how gender is distributed across the different professions in the study. I would stereotypically assume that healthcare workers are predominantly female, while police officers are largely male, but this should be explicitly stated to provide a clearer picture of the sample composition.
  • The fact that the VR sessions took place in two different locations—the research team’s university and the participant’s institution—should be specified in the data coding process, as this could potentially be a factor influencing the results.
  • It is unclear how the 'perceived psychological impact of the COVID-19 pandemic' was measured. Was a standardized scale used? If so, how was it composed? From Table 2, it seems to be a single item, but this needs to be clarified in the methodology section to ensure transparency in how the construct was assessed.
  • I am unclear about the timing of the interventions. How long after the initial measurement was the training conducted, and when were the post-measurements taken? This information should be clarified to better understand the timeline and the potential effects of the intervention.
  • Why is the Patient Health Questionnaire (PHQ-4) being used in this study, given that the respondents are not patients but healthcare and frontline workers? It would be helpful to justify the use of this tool for non-patient populations or consider an alternative measurement more suited to the study sample.
  • If this is the item used to assess the psychological impact of COVID-19—'To what extent do you consider that your experiences during the COVID-19 pandemic (2020–2022) harmed your psychological well-being?'—there are several issues to address. First, it is a single item, which limits the depth of measurement. Second, it refers to a period that is quite distant in time, potentially affecting the relevance of the response. Third, the item does not specify that it relates to work-related experiences, meaning that personal factors, such as the death of a relative, could have influenced the response.
  • Similarly, the following item, which asks participants to compare their psychological health at the time of the pandemic with their current psychological health, is not particularly useful. This question does not allow us to identify the reasons behind any changes in psychological well-being.

Results

  • The authors report that 'The main effect of Occupation was also significant (F[3,50] = 7.182, p < 0.001), indicating that across both time points, one or more groups had higher or lower stress scores than others when averaged over both time periods.' However, it does not seem to be specified which particular groups reported higher or lower stress levels. It would be helpful to provide more detail on which groups showed these differences to clarify the interpretation of the results.
  • I honestly don't understand the theoretical or methodological logic behind the regression analysis. Why should these variables be considered predictors of stress within a causal framework? In my view, this analysis should be excluded, as it lacks a clear theoretical justification and a solid methodological foundation for establishing causal relationships.

Discussion

  • Given the lack of a strong theoretical framework, the results in the discussion are consequently merely restated and addressed in a somewhat simplistic manner. As a result, they do not contribute significantly to the existing literature on either COVID-19 or virtual reality. A more robust theoretical underpinning and a deeper analysis of the findings would strengthen the paper's contribution to the field.
  • Practical implications and future directions should be emphasized more.

The main limitation of this study is that, in my opinion, it doesn't make sense to hypothesize a causal relationship between the distress experienced during COVID-19 (five years ago) and a 10-minute training session conducted years later, which is not even COVID-specific. This approach seems fundamentally flawed. Additionally, the absence of a control group is an incredibly important limitation. Without a control group, it is impossible to determine whether the observed effects are truly due to the VR intervention or if they might be attributed to other factors, such as natural changes in stress levels over time or the placebo effect. The lack of a control group significantly weakens the study’s internal validity and its ability to draw meaningful conclusions about the intervention's effectiveness. Another critical limitation is the reliance on a single session of VR training and a single post-measurement. As it stands, the results only show that a VR session of this type leads to short-term stress reduction, without providing insights into long-term effects or the broader applicability of the intervention. I believe the paper should be revised in this direction—removing the COVID dimension and instead exploring the effect with a longer follow-up period, involving more adequately represented professional groups, and ensuring that these groups work in high-stress or burnout-prone sectors.

Author Response

We would like to thank the reviewers for their thoughtful and detailed comments and suggestions, which greatly contributed to improving our manuscript. A formal acknowledgment has been included in the revised version of the article to recognize the constructive feedback provided during the review process. Please find our detailed responses in the attached Word document, along with the corresponding revisions. All changes have been highlighted in yellow in both the revised manuscript and the updated highlights document.

Reviewer 2 Report

Comments and Suggestions for Authors

While the manuscript addresses a relevant and timely topic — the psychological impact of the COVID-19 pandemic on frontline workers and the use of virtual reality (VR) as a stress-reduction tool — there are several methodological limitations that may hinder its suitability for a high-impact international journal.

Specifically, the study lacks a control group, which limits internal validity and makes it difficult to attribute observed changes to the VR intervention itself. Additionally, the sample size is small (n = 54), and the subgroups are even smaller, reducing statistical power and generalizability. The study uses a quasi-experimental, non-randomized design, relies solely on self-report measures, and evaluates only the immediate short-term effects of a single intervention session, without any follow-up to assess durability.

To meet higher standards of rigor, a future version of this study should consider including a randomized control group, objective physiological measures, a larger and more diverse sample, and longitudinal data.

Author Response

(The authors gave the same response as above.)

Reviewer 3 Report

Comments and Suggestions for Authors

The study is very interesting because it deals with the well-being of emergency workers during the Covid-19 Pandemic. It was precisely during the pandemic period that more attention was given in the scientific field not only to the victims but also to rescue and healthcare workers.

I have some suggestions to offer authors.

1) the variables that are the object of the study are indicated in the abstract, such as well-being, perceived stress. some variables have the indication of the instrument used, others do not. it would be appropriate to standardize, indicating the relative instrument for all the variables (for example resilience does not have its own instrument) or leave the variable more generally

2)The authors have cited many studies and the list of references is very rich. However, I suggest considering how emergency workers have been exposed to the risk of much more immediate symptoms of depression, such as secondary trauma (see Giostra et al. 2025) or emotional exhaustion as a component of burnout (see Bartone's studies during covid). These studies also considered how frontline workers were exposed to greater risks and developed more severe symptoms and how resilience played a protective role

3) the authors wrote "VR has been successfully applied to treat or alleviate 95
symptoms of anxiety disorders, depression, and post-traumatic stress by providing immersive relaxation, distraction from real-world stressors, or virtual exposure therapy in a controlled setting" these sentences seem to be in line with the studies by Giostra and Bartone indicated above where it was demonstrated that the Stop Negative thoughts-emotions strategy allowed emergency workers not to be overwhelmed by death anxiety and stress. This evidence, also in light of the results obtained by the authors in their study, could be highlighted in the discussion and/or conclusions to give more scientific support to their results. suggesting practical  tools for emergency workers to stop negative thoughts and emotions during emergencies seems very useful.

4) the first of the aims (line 119) seems too generic and already known in literature. It could be useful to make it more specific

5) there are some main critical points:
a) the data collection took place years after the end of the pandemic and it is not possible to establish whether the psychological state of the participants was affected by other work events
b) what control measures for the reliability of the data were taken? the tests used do not have control items or subscales
c) the psychological state of the rescue workers, as demonstrated by the studies, was different in the acute phases compared to the last phases of covid-19, especially after vaccination

d) the data collected cannot be retroactive and cannot provide precise indications on the effectiveness of the VR technique during or immediately after a health emergency.

e) Was the effect of the administration not measured? If filling out the questionnaires before or after the RV session could vary the results?

f) Asking for the perception of one's well-being 4-5 years later may be a random perception modified by cognitive, social and contextual factors both increasing and decreasing. This appears to be a central limitation of the study. Independent factors subsequent to the pandemic period may have altered the perception of one's well-being in that phase.

g) however only 2 participants requested psychological support and 27 participants declared an unchanged psychological state during the high spread of covid-19. (table 2) 12 participants improved. these results are in contrast with most studies and this could be linked to past. Emergency workers use during the emergency coping of denial of emotional states and negative thoughts. over time this denial may have led to a minimization of the psychological state

h) Why was resilience only measured in the pre-RV condition?

6) Table 1 is not necessary. They are descriptive data already reported in the sample description.

7) the results of table 4 can be inserted in table 3 avoiding repeating similar data. The means and SD of pre vs post RV are different from those presented in table 3. In table 3 no significant differences seem to emerge, while in table 4 the values ​​assume significance, albeit minimal.

8) to perform a repeated measures anova model a sufficient sample size is needed. it is not clear how many groups were considered and the description of the model should be made clearer. The g*power analysis suggests a sample size greater than 100 participants on 4 dependent variables and a fixed factor (pre vs post). This aspect must be clarified by the authors in order to consider the analysis as correct and feasible.

9)why do a repeated measures anova model and then a regression model? maybe it could be easier to do two regression models pre and post RV without anova. What the authors call model 1 and 2 in table 5 refers to the steps? It must be said that the effects recorded in table 5 are minimal and of low significance.

Author Response

(The authors gave the same response as above.)

Round 2

Reviewer 1 Report

Comments and Suggestions for Authors

Although the revisions made have improved the overall quality of the work, several critical issues remain unresolved. Specifically, concerns persist regarding the nature of the sample and the potential differences among participants (revise the nature of H2), the absence of a control group, and the type of measurement used to assess COVID-related stress. Additionally, the choice of analysis raises methodological questions.

I honestly do not understand the theoretical or methodological rationale behind the regression analysis employed. It is still unclear why the selected variables should be considered predictors of stress within a causal framework.

In conclusion, while the theoretical structure of the paper has improved, key weaknesses remain. Moreover, the authors have not followed my previous recommendation to revise the focus of the study. I continue to believe that the paper would benefit from removing the COVID dimension and instead exploring the effect over a longer follow-up period, involving better-represented professional groups, particularly those working in high-stress or burnout-prone sectors.

Author Response

We once again thank the reviewers for their thoughtful, detailed comments and suggestions, which have greatly improved our manuscript. A formal acknowledgement has been added to the revised article to recognise the constructive feedback received during the review process. Please find our point-by-point responses in the attached Word document; all changes are highlighted in yellow in the revised manuscript.

Reviewer 3 Report

Comments and Suggestions for Authors

I thank the authors for the revision work they have done on their manuscript following the suggestions provided which had the purpose of improving their study. The changes made make the text clear and complete

Author Response

(The authors gave the same response as above.)
